# Genetic Variations Associated with Drug Resistance Markers in Asymptomatic *Plasmodium falciparum* Infections in Myanmar

**DOI:** 10.3390/genes10090692

**Published:** 2019-09-09

**Authors:** Yan Zhao, Ziling Liu, Myat Thu Soe, Lin Wang, Than Naing Soe, Huanping Wei, Aye Than, Pyae Linn Aung, Yuling Li, Xuexing Zhang, Yubing Hu, Haichao Wei, Yangminghui Zhang, Jessica Burgess, Faiza A. Siddiqui, Lynette Menezes, Qinghui Wang, Myat Phone Kyaw, Yaming Cao, Liwang Cui

**Affiliations:** 1Department of Immunology, College of Basic Medical Sciences, China Medical University, Shenyang 110122, China (Y.Z.) (Z.L.) (L.W.) (H.W.) (Y.L.) (X.Z.) (Y.H.) (H.W.) (Y.Z.) (Q.W.); 2Myanmar Health Network Organization, Yangon 11211, Myanmar (M.T.S.) (A.T.) (P.L.A.); 3Department of Public Health, Ministry of Health and Sports, Nay Pyi Taw 15011, Myanmar; 4Department of Internal Medicine, Morsani College of Medicine, University of South Florida, 3720 Spectrum Boulevard, Tampa, FL 33612, USA (J.B.) (F.A.S.) (L.M.) (L.C.)

**Keywords:** asymptomatic infection, *Plasmodium falciparum*, drug resistance genes, haplotypes, multidrug resistance

## Abstract

The emergence and spread of drug resistance is a problem hindering malaria elimination in Southeast Asia. In this study, genetic variations in drug resistance markers of *Plasmodium falciparum* were determined in parasites from asymptomatic populations located in three geographically dispersed townships of Myanmar by PCR and sequencing. Mutations in dihydrofolate reductase (*pfdhfr*), dihydropteroate synthase (*pfdhps*), chloroquine resistance transporter (*pfcrt*), multidrug resistance protein 1 (*pfmdr1*), multidrug resistance-associated protein 1 (*pfmrp1*), and Kelch protein 13 (*k13*) were present in 92.3%, 97.6%, 84.0%, 98.8%, and 68.3% of the parasites, respectively. The pfcrt K76T, pfmdr1 N86Y, pfmdr1 I185K, and *pfmrp1* I876V mutations were present in 82.7%, 2.5%, 87.5%, and 59.8% isolates, respectively. The most prevalent haplotypes for pfdhfr, pfdhps, pfcrt and pfmdr1 were 51I/59R/108N/164L, 436A/437G/540E/581A, 74I/75E/76T/220S/271E/326N/356T/371I, and 86N/130E/184Y/185K/1225V, respectively. In addition, 57 isolates had three different point mutations (K191T, F446I, and P574L) and three types of N-terminal insertions (N, NN, NNN) in the *k13* gene. In total, 43 distinct haplotypes potentially associated with multidrug resistance were identified. These findings demonstrate a high prevalence of multidrug-resistant *P. falciparum* in asymptomatic infections from diverse townships in Myanmar, emphasizing the importance of targeting asymptomatic infections to prevent the spread of drug-resistant *P.*
*falciparum*.

## 1. Introduction

The emergence and spread of parasites resistant to antimalarial drugs and mosquitoes resistant to insecticides threaten the recent gains in malaria control and challenge the goal for malaria elimination in the Greater Mekong Subregion (GMS) of Southeast Asia [1]. Antimalarial drug resistance in *P. falciparum* tends to emerge in low-transmission settings, particularly in Southeast Asia and South America, before expanding to high-transmission settings in sub-Saharan Africa [2]. Resistance to chloroquine (CQ) and later to sulfadoxine-pyrimethamine (SP) has been responsible for the increased mortality in African children [3,4]. Artemisinin (ART)-resistant *P. falciparum* parasites were first reported in Cambodia and subsequently in all countries of the GMS [5]. This has accelerated resistance development in parasites to artemisinin combination therapy (ACT) partner drugs, resulting in increased treatment failure rates with dihydroartemisinin-piperaquine (DP) in Cambodia and with artesunate-mefloquine (AS-MQ) in Cambodia and on the Thai-Myanmar border [6,7]. The timely monitoring of antimalarial resistance spread is essential for maintaining the recent progress in malaria control and for achieving the goal of regional malaria elimination.

As a high malaria-burden country within the GMS, Myanmar uses several first-line treatments for *P. falciparum* infections, including artemether-lumefantrine, AS-MQ, and DP. However, delayed clearance has been observed in all the three first-line ACTs [8]. Similar to other malaria-endemic areas, Myanmar experiences a large proportion of asymptomatic malaria infections, as demonstrated by active case surveillance of malaria infections among healthy populations [9,10]. In the Shwegyin township of southern Myanmar, mutations in genes associated with drug resistance including those with ART resistance were identified in asymptomatic infections [11]. Asymptomatic or sub-clinical malaria infections seldom cause acute disease, but they are capable of infecting mosquitoes, thus serving as silent reservoirs for continued malaria transmission. Therefore, prevalent antimalarial drug resistant strains in asymptomatic infections may underlie the rapid spread of drug resistance. Given the limited data on drug resistance in asymptomatic infections, ongoing molecular epidemiological studies of drug resistance are needed in these parasite reservoirs.

Molecular epidemiology studies provide information for detecting the emergence and tracking the spread of antimalarial drug resistance. Several mutations in the *P. falciparum* dihydrofolate reductase (*pfdhfr*) and *P. falciparum* dihydropteroate synthase (*pfdhps*) genes (e.g., triple mutations at codons 51, 59 and 108 of *pfdhfr* and double mutations at codons 437 and 540 of *pfdhps*) are associated with SP treatment failures [12]. The *P. falciparum* chloroquine resistance transporter (*pfcrt*) K76T mutation and *P. falciparum* multidrug resistance protein 1 (*pfmdr1*) N86Y mutation have been linked to CQ and amodiaquine resistance [13]. Mutations in the *P. falciparum* multidrug resistance-associated protein 1 (*pfmrp1*), such as H191Y and S437A, were reported to be associated with resistance to CQ and quinine (QN) in vitro [14]. Point mutations in the propeller domain of Kelch protein 13 (*k13*) were collectively correlated with clinical ART resistance [15,16]. Molecular surveillance showed that *k13* mutations associated with ART resistance were restricted to certain areas of the GMS, with C580Y being the predominant mutation in the Thai-Cambodian region and F446I in China and northern Myanmar [17,18,19,20].

Here, we profiled mutations in antimalarial drug resistance genes in asymptomatic *P. falciparum* infections identified from cross-sectional studies in three townships of Myanmar. The results will provide updated information on the drug resistance status of *P. falciparum* malaria in multiple sentinel sites in Myanmar, which is necessary for developing strategies to eliminate *P. falciparum* infections in this country.

## 2. Materials and Methods

### 2.1. Ethics Approval

Ethical approval for this project was obtained from the institutional review boards of China Medical University, China (2019086), University of South Florida, USA (Pro00036813) and the Ministry of Health and Sports, Myanmar (Ethics/DMR/2017/077AE/2018). Before conducting the study, all adult participants or legal guardians of children voluntarily signed the informed consent.

### 2.2. Study Sites and Samples

Three cross-sectional studies were conducted in Myanmar at the Laiza (Kachin State), Banmauk (Sagaing Region), and Paletwa (Chin State) townships in 2015, 2017–2018, and 2017, respectively (Figure 1A). From 2015 to 2017, a country profile of malaria showed that Paletwa had the highest annual incidence of *P. falciparum*, followed by Banmauk and Laiza [21,22,23]. These cross-sectional surveys recruited a total of 3,495 residents, who provided finger-prick blood to prepare blood smears and dried blood spots on filter paper (Figure 1B).

### 2.3. Diagnosis of *P. falciparum* Asymptomatic Infections

Smears were stained with Giemsa and examined by two experienced microscopists. For molecular diagnosis, parasite genomic DNA was isolated from blood spots on filter paper according to the protocol of QIAamp^®^ DNA Mini kit (Qiagen). Detection of *P. falciparum* isolates was performed by nested PCR using an established protocol targeting the 18S rRNA gene as described previously [24]. This method has a detection limit of ~2 parasites/μL. For PCR assessment, one positive and one negative control (from a symptomatic *P. falciparum* isolate and sterile water, respectively) were used in each of the amplification plates. The genomic DNA from *P. falciparum* infections was used for the amplification of genes associated with drug resistance.

### 2.4. PCR and Sequencing of Drug Resistance Markers

The target sequences covering the resistance-conferring mutations from each gene were selected for PCR amplification and sequencing using published protocols [15,25,26,27,28]. The sample DNA typically had OD260/280 between 1.6 and 1.8 with a concentration of >5 ng/µL as measured using the NanoDrop 2000C spectrophotometer. PCR amplicons were purified and sequenced using an ABI 3730XL DNA analyzer. For sequence accuracy, all DNA fragments were sequenced for both strands. The primers used to amplify these genes are summarized in Appendix A and S2. These selected regions include codons 51, 59, 108, and 164 of *pfdhfr* (PF3D7_0417200); 436, 437, 540, and 581 codons of *pfdhps* (PF3D7_0810800); and three *pfmdr1* (PF3D7_0523000) fragments including codons 86, 184, 1034, 1042, and 1246. For *pfcrt* (PF3D7_0709000), *pfmrp1* (PF3D7_0112200) and *k13* (PF3D7_1343700), full-length genes were amplified and sequenced (Appendix A). The gene sequences for *k13*, *pfdhfr*, *pfdhps*, *pfmdr1*, *pfmrp1* and *pfcrt* reported in this study were deposited in GenBank under accession numbers MN419439 - MN419894.

### 2.5. Sequence Analysis and Statistics

We evaluated the quality of sequences by examining the chromatograms. Those showing ambiguity were re-amplified and re-sequenced. If the re-sequencing did not improve the quality, these samples were excluded from the analysis. The sequences were aligned using ClustalW in MEGA7.0.26. The samples showing mixed chromatograms were excluded and only single-peak sequences were considered to be monoclonal *P. falciparum* infections and were used for single nucleotide polymorphism (SNP) and haplotype analysis (Appendix A). Nucleotide and amino acid positions were numbered according to the 3D7 reference sequences. The SNP data from different genes were analyzed using Microsoft Soft Excel 2007 and SPSS 22.0. Pearson’s Chi-square test or Fisher’s exact test were used to determine statistical significance (*p* < 0.05). The haplotype network was constructed by DnaSP 6 and Network software 5.01.0 using the median joining algorithm [29].

### 2.6. Availability of Data and Material

The datasets used and/or analyzed during the current study are available from the corresponding authors upon request.

## 3. Results

### 3.1. Asymptomatic Infections in Blood Samples from Cross-Sectional Surveys

Asymptomatic *P. falciparum* infections were prevalent in malaria-endemic areas of Myanmar [24,30]. Dried blood samples on filter papers from 3495 healthy residents collected in cross-sectional surveys in three regions of Myanmar were screened by nested PCR. A total of 107 asymptomatic *P. falciparum* infections were identified for an overall prevalence of 3.1%. After excluding the samples with double peaks suggestive of mixed infections, 80 *pfdhfr*, 82 *pfdhps*, 75 *pfcrt*, 80 *pfmdr1*, 82 *pfmrp1*, and 57 *k13* sequences were used for SNP and haplotype analysis (Figure 1B).

### 3.2. *pfdhfr* and *pfdhps* Mutations Associated with Antifolate Resistance

Mutations in *pfdhfr* at codons 51, 59, 108, and 164 were present in 60.0%, 90.0%, 92.5%, and 40.0% samples, respectively (Table 1). The majority of the isolates carried mutations in *pfdhfr* at codon 59 and 108. In Laiza, 100.0% (12/12) of the isolates had all the four mutations except one isolate lacking the N51I mutation. In Banmauk and Paletwa, the S108N mutation was predominant at 88.1% (37/42) and 96.2% (25/26), respectively. Analysis of *pfdhfr* haplotypes revealed that the quadruple mutant 51I/59R/108N/164L was predominant and showed significant differences in prevalence among the three townships; the highest in Laiza at 91.7% (*p* < 0.0001; Figure 2A and Table 1). The triple mutant haplotype 51I/59R/108N was predominantly in Banmauk (15/42; 35.7%), whereas the double mutant 59R/108N was more prevalent in Paletwa (Table 1). This suggests a trend of decreased resistance to pyrimethamine from the east to the west borders of Myanmar.

For the *pfdhps* gene, mutations at codons 436, 540, and 581 were present in Laiza and Paletwa, in 25.0% vs. 55.6%, 100.0% vs. 92.6% and 75.0% vs. 25.9% samples, respectively (Table 2). In Banmauk, isolates carrying mutations at codons 436, 437, 540, and 581 were at 20.9%, 48.8%, 32.5%, and 25.6%, respectively. Importantly, in addition to the point mutation K540E, the K540N variant was found in Laiza and Banmauk for the first time. Comparison of the *pfdhps* haplotypes revealed significant differences among the three townships. In Laiza, 41.7% (5/12) and 33.3% (4/12) of the isolates carried the double mutant 540N/581G and the variant 540E/581G, respectively, whereas in Banmauk the most common haplotype carried the single mutation 437A at 46.5% (20/43) (Table 2). Overall, the double mutant 436A/540E was most dominant and differed significantly in frequency among the three townships with the highest prevalence in Paletwa at 55.6% (*p* = 0.003; Figure 2B and Table 2).

### 3.3. *Pfcrt* and *pfmdr1* Mutations Associated with CQ Resistance

*Pfcrt* gene was successfully sequenced in 75/106 (70.8%) of the isolates, with clear single peaks covering the amino acids 74, 75, 76, 220, 271, 326, 356, and 371. Except for N326S, all these mutations were highly prevalent, ranging from 78.7% to 84.0% (Table 3). The septuple mutant was the most prevalent haplotype (52.0%, 39/75) (Figure 2C), and it was observed in 75.0% of the isolates in Paletwa, 53.8% in Banmauk, but none in Laiza (*p* < 0.001). In Laiza, 100% of parasites carried mutations at all these amino acids, and the octuple mutant was also present at lower frequencies in Banmauk (15.4%) and Paletwa (8.3 %), respectively (*p* < 0.001). The sextuple mutant was present exclusively in three isolates from Paletwa (*p* = 0.057).

*Pfmdr1* mutations at codons 86, 130, 184, 185, and 1225 were observed in this study (Table 4). The I185K mutation has been rarely reported previously, but it was the dominant mutation (87.5%) in our samples. In comparison, other mutations were found at lower frequencies, ranging from 1.3% to 23.8%. The I185K and the Y184F mutations differed significantly in frequency among the three townships (*p* < 0.001 & *p* = 0.002 respectively). N86Y, a mutation associated with CQ resistance, was only found in Paletwa (7.4%, 2/27). The prevalence of *pfmdr1* haplotypes varied in different study sites. The single mutant (b) was the most frequent haplotype in Banmauk (80.5%, 33/41) and Paletwa (74.1%, 20/27), respectively. However, the single mutant (a) was the predominant haplotype in Laiza (58.3%, 7/12). Notably, one isolate from Paletwa carried the *pfmdr1* N86Y mutation along with the *pfcrt* K76T mutation.

### 3.4. *Pfmrp1* Mutations

For the *pfmrp1* gene, the mutations at positions 191, 325, 437, 572, 785, 876, 1007, 1339, and 1390 were detected in 65.9%, 2.4%, 65.9%, 2.4%, 22.0%, 59.8%, 50.0%, 2.4% and 8.5% of the asymptomatic *P. falciparum* isolates, respectively (Table 5). The prevalence of mutations H191Y, S437A, H785N, I876V, and T1007M was significantly different among the three townships (*p* = 0.002, *p* = 0.002, *p* < 0.001, *p* = 0.029 and *p* < 0.001, respectively). Ten distinct haplotypes were detected in *pfmrp1*, with the wild-type (WT) being the most prevalent haplotype, present in 31.7% of the isolates (Figure 2E and Table 5). The quintuple mutant (a) at 50.0% and quadruple mutant (b) at 77.8% were the predominant haplotypes in Laiza and Paletwa, respectively, whereas 46.5% of the isolates had the WT haplotype in Banmauk (Table 5).

### 3.5. *K13* Mutations Associated with ART Resistance

The *k13* gene was successfully sequenced in 57 samples. Three different point mutations were found, including two previously reported mutations (F446I and P574L) and one mutation (K191T) observed for the first time in these samples (Table 6). Of these mutations, F446I was most frequent (14.0%; 8/57). Besides point mutations, N, NN, and NNN insertions between amino acids 136 and 137 were also identified in 1 (1.8%), 26 (45.6%) and 4 (7.0%) samples, respectively. All *P. falciparum* isolates from Laiza had the NN insert, whereas in Paletwa and Banmauk, it was present in 40.0% and 33.3% of the samples, respectively (*p* < 0.001). The F446I mutation always occurred together with the NN insert and was present exclusively in samples from Laiza along the China-Myanmar border. Of the 57 asymptomatic *P. falciparum* isolates, 22 samples were WT (Figure 2F and Table 6).

### 3.6. Haplotype Network

SNPs in the six genotyped drug-resistance genes gave rise to 43 distinct haplotypes in 52 *P. falciparum* isolates that were successfully sequenced at all six target genes (Appendix A), demonstrating high-level genetic diversity of asymptomatic *P. falciparum* infections in these regions. Among the samples, 11.6% (5/52), 7.0% (3/52), 7.0% (3/52), and 4.7% (2/52) shared haplotypes No. 2, 9, 30, and 26, respectively, whereas each of the remaining isolates was unique. To examine the phylogenetic relationship of the asymptomatic *P. falciparum* parasites, a haplotype network was generated based on the SNPs observed in the six genes (Figure 3). The network had three main branches. Isolates of the three study areas appeared to be polyphyletic and were distributed across the entire network, suggesting independent origins of resistant mutations in different genes from diverse genetic backgrounds. Three haplotypes with the *k13* K191T mutation originated from Banmauk. Four haplotypes with the *k13* F446I mutation and one haplotype with the P574L mutation associated with ART resistance appeared independently in Laiza. Significantly, there were 20 haplotypes with the NN insert in *k13* gene, including eight haplotypes in Laiza, six in Banmauk and six in Paletwa.

## 4. Discussion

As the GMS is moving towards malaria elimination, Myanmar requires immediate attention because of the high malaria burden and its geographical connection with South Asia, through which drug resistance could spread rapidly. According to WHO reports, the latest malaria drug resistance map reveals that both treatment failure and resistant genotypes are prevalent in Myanmar, including our three study sites [31]. Irrespective of the ACTs used, from 2010 to 2018, treatment failure among *P. falciparum* patients in Kachin State was higher than that in Chin State. Our study found a higher prevalence of *k13* and *pfmdr1* mutations in Kachin State (east) than in Chin State (west) which is possibly contributing to the treatment failure in Kachin State. Asymptomatic *Plasmodium* infections, as silent reservoirs of malaria parasites, play an important role in continued malaria transmission. Critically, these asymptomatic infections may carry genetic variations conferring drug resistance, thus facilitating the spread of drug-resistant parasites [11]. Therefore, this study aimed to provide molecular epidemiology information about drug-resistant *P. falciparum* in asymptomatic infections at three sentinel sites located in southwestern, northern and northeastern Myanmar, each with a different malaria epidemiology.

The drug combination SP, commercially known as Fansidar, has been used as an antimalarial chemotherapy in Southeast Asia and Africa. The mechanism of resistance to sulfadoxine has been associated with five point mutations at the S436A/F, G437A, K540E, A581G, and A613T/S codons of the *pfdhps* gene. Mutations at 437 and 540 were strongly associated with SP treatment failure, whereas the 436, 581, and 613 mutations confer some degrees of resistance [32]. In this study, A613T/S was not observed, but K540N was prevalent, consistent with a previous study from northeast Myanmar [33]. A581G, G437A, and K540E were the predominant mutations in Laiza, Banmauk, and Paletwa, respectively. In population studies, mutations at codon 59 of the *pfdhfr* gene and codon 540 of the *pfdhps* gene are strongly predictive of SP treatment failure [34,35]. The prevalence of this mutation combination in Laiza and Paletwa was 100% and 83.3%, respectively, and was slightly lower in Banmauk (34.1%), suggesting that strong selection pressure against SP was emerging in western and northeastern Myanmar. Similarly, triple mutations at codons 108, 51, and 59 of *pfdhfr* and double mutations at codons 437 and 540 of *pfdhps* are associated with SP treatment failure [12]. Here, one isolate was found to have this quintuple mutation combination in Banmauk, and 12 isolates (nine in Laiza, two in Banmauk and one in Paletwa) harbored the sextuple mutations with *pfdhfr* (N51I, C59R, S108N, and I164L) and *pfdhps* (K540N/E and A581G), suggesting that *P. falciparum* from these asymptomatic carriers would be highly resistant to SP.

Mutations in the *pfcrt* gene, especially the K76T mutation, are major determinants of CQ resistance [36,37]. In addition, *pfcrt* mutations also influence *P. falciparum* susceptibility to mefloquine (MQ), halofantrine (HF), and ART [38]. The K76T mutation is associated with different sets of mutations at different codons, most commonly C72S, M74I, N75E, A220S, Q271E, N326S, I356T, and R371I [39]. Consistent with a previous study conducted in Thailand [25], C72S was not found. Except for N326S, other mutations reached high prevalence in all three sites. Notably, almost all parasite isolates had the CIVIET haplotype around codons 72-76, which is the typical CQ-resistant haplotype in Southeast Asia. The stable and highly prevalent of *pfcrt* mutations may be the result of continued use of CQ as the first-line treatment for *P. vivax* infections in this region [40,41]. In addition to CQ resistance, *pfcrt* has gained recognition as a multidrug resistance transporter, which influences parasites’ susceptibilities to multiple first-line antimalarial drugs [42,43]. Emerging mutations in *pfcrt*, H97Y, F145I, M343L, and G353V, also have been linked to piperaquine (PPQ) resistance [44,45], but these mutations were not identified in this study. Thus, the high prevalence of *pfcrt* mutations in asymptomatic *P. falciparum* parasites may be linked to failures of multiple ACTs in Myanmar.

The ATP-binding cassette (ABC) transporters, including *pfmdr1* and *pfmrp1*, are potentially involved in resistance to multiple antimalarial drugs [46,47]. Several point mutations in *pfmdr1* are associated with changed sensitivities to CQ, MQ, QN, HF, and ART [38,48,49,50]. In the presence of *pfcrt* K76T, point mutations in *pfmdr1*, primarily at codon 86 [13,14] and additionally at positions 184, 1034, 1042, and 1246 [51], can increase resistance of the parasites to CQ. Decreased susceptibility to lumefantrine (LMF) has also been linked to polymorphism in *pfcrt* and *pfmdr1* [52,53]. In this study, only two *P. falciparum* isolates from the western township Paletwa had the *pfcrt* K76T and *pfmdr1* N86Y mutations, but the combination of *pfcrt* K76T and *pfmdr1* Y184F had much higher prevalence in Laiza (58.3%) than Banmauk (18.0%) and Paletwa (12.5%). The *pfmdr1* Y184F was not detected in an earlier study [54], where I185K was reported as a novel mutation. The authors speculated that the decrease in parasite’s susceptibility to the drug was associated with substitution of mutation at codon Y184F for I185K. In our study, the *pfmdr1* I185K mutation was highly prevalent (~90% in all samples) and reached fixation in Banmauk and Paletwa. It would be worthwhile to establish whether this mutation is linked to altered drug sensitivity through genetic studies given the high prevalence of this mutation. Since disruption of *pfmrp1* in a CQ-resistant *P. falciparum* isolate rendered this parasite more sensitive to CQ, QN, PPQ, primaquine, and ART, PfMRP1 is proposed to mediate resistance of the parasite to multiple antimalarial drugs [55]. Association studies suggest that mutations at codons Y191H, A437S, H785N, I876V and F1390I in *pfmrp1* contribute to decreased sensitivity to CQ, PPQ, LMF, artesunate (ATS) and dihydroartemisinin (DHA) as well as reduce both in vivo and in vitro susceptibility to ACT [28,33,56]. Our results suggest that *P. falciparum* isolates from asymptomatic infections in these areas might be resistant to CQ, PPQ, LMF, ATS, DHA, and ACT.

ART resistance in *P. falciparum* has emerged in western Cambodia and its potential spread to Africa threatens global malaria control [57]. ART resistance is characterized by delayed parasite clearance, which corresponds to the decreased susceptibility of ring-stage parasites [58]. Non-synonymous SNPs in K13 were found to be strongly associated with resistance to ART [16]. To date, more than 200 non-synonymous mutations in the *k13* gene have been reported. N458Y, Y493H, R539T, I543T, and C580Y are validated ART resistance mutations, and a number of mutations are candidates for ART resistance, including P574L [8]. In Africa, non-synonymous *k13* mutations are relatively rare [59]. Recently, *k13* mutations have emerged independently in multiple locations in Southeast Asia, including Myanmar [17]. C580Y, Y493H, R539T, and I543T *k13* mutations were prevalent in the Thai-Myanmar border, whereas F446I was the most prevalent mutant allele of *k13* in the northern Myanmar and China-Myanmar border, and P574L next to C580Y were reported among isolates collected from migrant goldmine workers in southern Myanmar [19,60,61,62]. In our study, the validated mutations of *k13* resistance were not found, while F446I, P574L, and K191T mutations in the *k13* propeller domain were observed among parasites from asymptomatic infections. The K191T has not been reported in previous studies. F446I mutation was observed with high prevalence only at Laiza along China-Myanmar border, which is consistent with previous studies. F446I mutation in *k13* has been found to be associated with delayed parasite clearance [61], and a recent study showed that F446I was associated with the increased ring survival rates by genetically introducing the mutation into the *k13* gene [63]. Consistent with previous studies of *P. falciparum* from a symptomatic population, the absence of *k13* propeller mutations in Paletwa suggest that ART resistant parasites may not have spread to the western Myanmar region [18]. Moreover, we found the N, NN, and NNN insertion in the N terminus of *k13* protein through analysis of the full-length *k13* sequences. A previous study reported that an NN insertion in the N terminus of *k13* protein was associated with increased in vitro ring-stage survival in the presence of DHA, but the possibility of the NN insert acting cooperatively with F446I in conferring ART resistance was not ruled out [33]. In view of the high prevalence (45.6%) of the NN insertion among *P. falciparum* from asymptomatic populations in Myanmar, it would be valuable to determine whether it independently affects ART resistance.

The generation of the haplotype network can be used to study the phylogenetic relationship of parasites, to determine the origins of drug resistance gene mutations from different geographical regions, and to assess the diversity of parasite populations. Consistent with the results from clinical *P. falciparum* isolates, parasites from asymptomatic individuals also displayed extraordinarily high diversity in Myanmar. The genotypes of the six genes associated with drug resistance indicated that the asymptomatic *P. falciparum* infections in Myanmar are resistant to multiple drugs. More importantly, 58.8% (20/34) haplotypes carried *k13* mutations (N458Y, R539T, P574L, and F446I) associated with ART resistance in clinical isolates, whereas 62.5% (5/8) haplotypes of asymptomatic *P. falciparum* isolates carried F446I and P574L. Although only two *k13* mutations related to ART resistance were detected in our study, the presence of such high haplotype diversity in the limited isolates of asymptomatic infection should be concerning. Despite reduced *P. falciparum* prevalence in recent years, the evidence suggests that ART resistant *P. falciparum* parasites appear independently and spread widely along the China-Myanmar border, which is a reminder that surveillance of *P. falciparum* should be strengthened both in Myanmar and China. Although *k13* mutation associated with ART resistance was not found in the other two areas, the diversity of multiple drug resistance reflected by haplotype analysis should not be ignored.

## 5. Conclusions

In this study, analysis of six molecular markers of drug resistance among asymptomatic *P. falciparum* infections revealed a high prevalence of mutations linked to drug resistance in Myanmar. High mutation frequency in *pfdhfr*, *pfdhps*, and *pfcrt* means a serious resistance to SP and CQ in these areas, which supports the ACT being used for *P. falciparum* treatment in Myanmar. Fortunately, ART resistance has not yet spread to western Myanmar. Given that *pfcrt* also is a multidrug resistance transporter with high frequency of mutations, it is necessary to continuously monitor for the emergence of ART resistance in western Myanmar. However, the drug resistance pattern of *P. falciparum* from asymptomatic carriers along the China-Myanmar border is not optimistic; perhaps the best option to prevent the spread of drug-resistant parasites in border areas is rapid elimination through mass drug administration. Haplotype analysis also revealed extremely high diversity in these areas. These asymptomatic *P. falciparum* infections carrying mutations associated with drug resistance may be an important reservoir for the transmission and spread of resistance in this region. Molecular surveillance of antimalarial resistance might be helpful for developing and updating guidance for the use of antimalarials in Myanmar.

## Figures and Tables

**Figure 1 genes-10-00692-f001:**
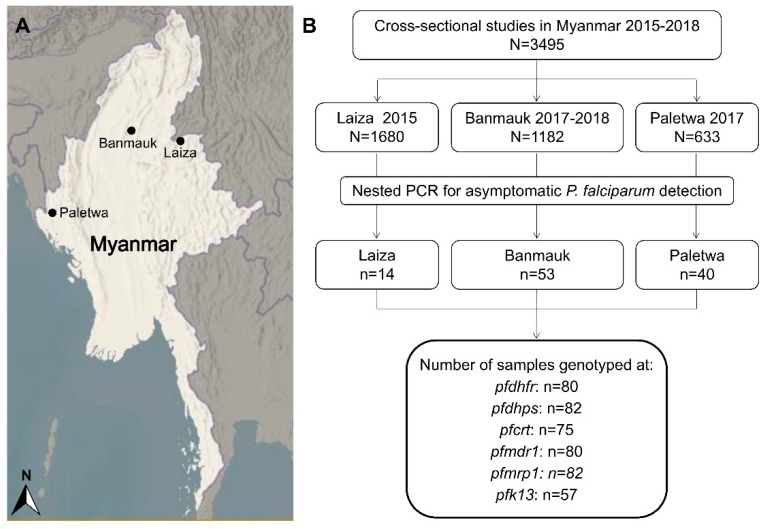
Map of study areas and flow chart of resistance markers detection. (**A**) Three study townships are indicated as black dots in the maps. (**B**) The sequences of the samples in the final black coiled frame were analyzed for single nucleotide polymorphism (SNP) and haplotypes.

**Figure 2 genes-10-00692-f002:**
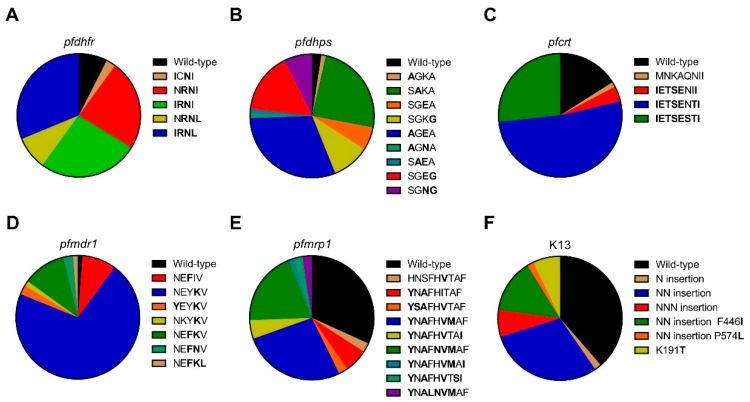
Prevalence of dihydrofolate reductase (*pfdhfr*), dihydropteroate synthase (*pfdhps*), chloroquine resistance transporter (*pfcrt*), multidrug resistance protein 1 (*pfmdr1*), multidrug resistance-associated protein 1 (*pfmrp1*), and Kelch protein 13 (*k13*) haplotypes in *P. falciparum* isolates from asymptomatic carriers. (**A**) Prevalence of *pfdhfr* haplotypes, mutations at codons 51, 59, 108, and 164. including wild-type NCSI, double mutant **I**C**N**I and N**RN**I, triple mutant **IRN**I and N**RNL**, and quadruple mutant **IRNL**; (**B**) Prevalence of *pfdhps* haplotypes, mutations at codons 436, 437, 540, and 581, including wild-type SGKA, single mutant **A**GKA, S**A**KA, SG**E**A, and SGK**G**, double mutant **A**G**E**A, **A**G**N**A, S**AE**A, SG**EG**, and SG**EG**; (**C**) Prevalence of *pfcrt* haplotypes, mutations at codons 74, 75, 76, 220, 271, 326, 356, and 371, including wild-type MNKAQNIR, single mutant MNKAQNI**I**, sextuple mutant **IETSE**NI**I**, septuple mutant **IETSE**N**TI**, and octuple mutant **IETSESTI**; (**D**) Prevalence of *pfmdr1* haplotypes, mutations at codons 86, 130, 184, 185, and 1225, including wild-type NEYIV, single mutant NE**F**IV and NEY**K**V, double mutant **Y**EY**K**V, N**K**Y**K**V, NE**FK**V, and NE**FN**V, triple mutant NE**FKL**; (**E**) Prevalence of *pfmrp1* haplotypes, mutations at codons 191, 325, 437, 572, 785, 876, 1007, 1339, and 1390, including wild-type HNSFHITAF, single mutant HNSFH**V**TAF, double mutant **Y**N**A**FHITAF, quadruple mutant **YSA**FH**V**TAF, **Y**N**A**FH**VM**AF, and **Y**N**A**FH**V**TA**I**, quintuple mutant **Y**N**A**F**NVM**AF, **Y**N**A**FH**VM**A**I**, and **Y**N**A**FH**V**T**SI**, sextuple mutant **Y**N**ALNVM**AF; (**F**) Prevalence of *k13* haplotypes, mutations at codons 191, 446, 574 and N, NN, NNN insertion, including wild-type, N insertion, NN insertion, NNN insertion, NN insertion with 446I, NN insertion with 574L, and single mutant 191T.

**Figure 3 genes-10-00692-f003:**
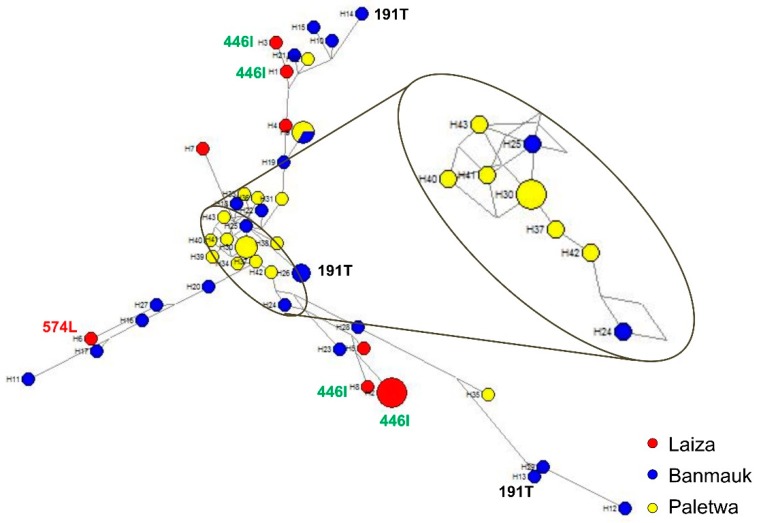
Median-joining haplotype network of asymptomatic *P. falciparum* isolates harboring mutations in six drug resistance-associated genes. The haplotype network was constructed for asymptomatic *P. falciparum* isolates using 43 haplotypes obtained from amino acid changes observed in *pfdhfr*, *pfdhps*, *pfcrt*, *pfmdr1*, *pfmrp1*, and *k13*. The size of each circle shows the same haplotype prevalence, with a color corresponding to a population of origin: Laiza (red), Banmauk (blue), or Paletwa (yellow). The length of an edge is proportional to the number of variations between two haplotypes, and enlarged is the torso of the tree. K13 individual mutations are labeled by amino acid position.

**Table 1 genes-10-00692-t001:** Prevalence of dihydrofolate reductase *(pfdhfr*) mutations and haplotypes in asymptomatic *P. falciparum* isolates collected from three townships of Myanmar.

Mutations		Number of Isolate (%)				*p* Value ^#^
		Laiza	Banmauk	Paletwa	Total	
		n = 12	n = 42	n = 26	n = 80	
N51**I**		11 (91.7)	26 (61.9)	11 (42.3)	48 (60.0)	0.012
C59**R**		12 (100.0)	36 (85.7)	24 (92.3)	72 (90.0)	0.467
S108**N**		12 (100.0)	37 (88.1)	25 (96.2)	74 (92.5)	0.425
I164**L**		12 (100.0)	13 (31.0)	7 (26.9)	32 (40.0)	0.000
**Haplotype**	**Codon** *					
Wild-type	NCSI	0 (0.0)	5 (11.9)	1 (3.8)	6 (7.5)	0.425
Double mutant (a)	**I**C**N**I	0 (0.0)	1 (2.4)	1 (3.8)	2 (2.5)	1.000
Double mutant (b)	N**RN**I	0 (0.0)	8 (19.0)	11 (42.3)	19 (23.8)	0.009
Triple mutant (a)	**IRN**I	0 (0.0)	15 (35.7)	6 (23.1)	21 (26.3)	0.034
Triple mutant (b)	N**RNL**	1 (8.3)	3 (7.1)	3 (11.5)	7 (8.8)	0.863
Quadruple mutant	**IRNL**	11 (91.7)	10 (23.8)	4 (15.4)	25 (31.3)	0.000

* Point mutations are shown in boldface. ^#^ Comparison between the three different sites was done by Fisher’s exact test.

**Table 2 genes-10-00692-t002:** Prevalence of dihydropteroate synthase (*pfdhps*) mutations and haplotypes in asymptomatic *P. falciparum* isolates collected from three townships of Myanmar.

Mutation at Codon		Number of Isolate (%)				*p* Value ^#^
		Laiza	Banmauk	Paletwa	Total	
		n = 12	n = 43	n = 27	n = 82	
S436**A**		3 (25.0)	9 (20.9)	15 (55.6)	27 (32.9)	0.010
G437**A**		0 (0.0)	21 (48.8)	0 (0.0)	21 (25.6)	0.000
K540**E**		6 (50.0)	13 (30.2)	25 (92.6)	44 (53.7)	0.000
K540**N**		6 (50.0)	1 (2.3)	0 (0.0)	7 (8.5)	0.000
A581**G**		9 (75.0)	11 (25.6)	7 (25.9)	27 (32.9)	0.005
**Haplotype**	**Codon** *					
Wild-type	SGKA	0 (0.0)	2 (4.7)	0 (0.0)	2 (2.4)	0.650
Single mutant (a)	**A**GKA	0 (0.0)	1 (2.3)	0 (3.3)	1 (1.2)	1.000
Single mutant (b)	S**A**KA	0 (0.0)	20 (46.5)	0 (3.3)	20 (24.4)	0.000
Single mutant (c)	SG**E**A	0 (0.0)	0 (0.0)	5 (18.5)	5 (6.1)	0.007
Single mutant (d)	SGK**G**	0 (0.0)	6 (14.0)	2 (7.4)	8 (9.8)	0.405
Double mutant (a)	**A**G**E**A	2 (16.7)	8 (18.6)	15 (55.6)	25 (30.5)	0.003
Double mutant (b)	**A**G**N**A	1 (8.3)	0 (0.0)	0 (0.0)	1 (1.2)	0.146
Double mutant (c)	S**AE**A	0 (0.0)	1 (2.3)	0 (0.0)	1 (1.2)	1.000
Double mutant (d)	SG**EG**	4 (33.3)	4 (9.3)	5 (18.5)	13 (15.9)	0.094
Double mutant (e)	SG**NG**	5 (41.7)	1 (2.3)	0 (0.0)	6 (7.3)	0.000

* Point mutations are shown in boldface. ^#^ Comparison between the three different sites was done by Fisher’s exact test (*p* < 0.05).

**Table 3 genes-10-00692-t003:** Prevalence of chloroquine (CQ) resistance transporter (*pfcrt*) mutations and haplotypes in asymptomatic *P. falciparum* isolates collected from three townships of Myanmar.

Mutation at Codon		Number of Isolate (%)				*p* Value ^#^
		Laiza	Banmauk	Paletwa	Total	
		n = 12	n = 39	n = 24	n = 75	
M74**I**		12 (100.0)	27 (69.2)	23 (95.8)	62 (82.7)	0.005
N75**E**		12 (100.0)	27 (69.2)	23 (95.8)	62 (82.7)	0.005
K76**T**		12 (100.0)	27 (69.2)	23 (95.8)	62 (82.7)	0.005
A220**S**		12 (100.0)	27 (69.2)	23 (95.8)	62 (82.7)	0.005
Q271**E**		12 (100.0)	27 (69.2)	23 (95.8)	62 (82.7)	0.005
N326**S**		12 (100.0)	6 (15.4)	2 (8.3)	20 (26.7)	0.000
I356**T**		12 (100.0)	27 (69.2)	20 (83.3)	59 (78.7)	0.025
R371**I**		12 (100.0)	28 (71.8)	23 (95.8)	63 (84.0)	0.016
**Haplotype**	**Codon** *					
Wild-type	MNKAQNIR	0 (0.0)	11 (28.2)	1 (4.2)	12 (16.0)	0.011
Single mutant	MNKAQNI**I**	0 (0.0)	1 (2.6)	0 (0.0)	1 (1.3)	1.000
Sextuple mutant	**IETSE**NI**I**	0 (0.0)	0 (0.0)	3 (12.5)	3 (4.0)	0.057
Septuple mutant	**IETSE**N**TI**	0 (0.0)	21 (53.8)	18 (75.0)	39 (52.0)	0.000
Octuple mutant	**IETSESTI**	12 (100.0)	6 (15.4)	2 (8.3)	20 (26.7)	0.000

* Point mutations are shown in boldface with underline. ^#^ Comparison between the three different sites was done by Fisher’s exact test (*p* < 0.05).

**Table 4 genes-10-00692-t004:** Prevalence of multidrug resistance protein 1 (*pfmdr1*) mutations and haplotypes in asymptomatic *P. falciparum* isolates collected from three townships of Myanmar.

Mutation at Codon		Number of Isolate (%)				*p* Value #
		Laiza	Banmauk	Paletwa	Total	
		n = 12	n = 41	n = 27	n = 80	
N86**Y**		0 (0.0)	0 (0.0)	2 (7.4)	2 (2.5)	0.234
E130**K**		0 (0.0)	0 (0.0)	1 (3.7)	1 (1.3)	0.488
Y184**F**		7 (58.3)	8 (19.5)	4 (14.8)	19 (23.8)	0.014
I185**K**		4 (33.3)	39 (95.1)	27 (100.0)	70 (87.5)	0.000
I185**N**		0 (0.0)	2 (4.9)	0 (0.0)	2 (2.5)	0.650
V1225**L**		0 (0.0)	1 (2.4)	0 (0.0)	1 (1.3)	1.000
**Haplotype**	**Codon** *					
Wild-type	NEYIV	1 (8.3)	0 (0.0)	0 (0.0)	1 (1.3)	0.150
Single mutant (a)	NE**F**IV	7 (58.3)	0 (0.0)	0 (0.0)	7 (8.8)	0.000
Single mutant (b)	NEY**K**V	4 (33.3)	33 (80.5)	20 (74.1)	57 (71.3)	0.008
Double mutant (a)	**Y**EY**K**V	0 (0.0)	0 (0.0)	2 (7.4)	2 (2.5)	0.234
Double mutant (b)	N**K**Y**K**V	0 (0.0)	0 (0.0)	1 (3.7)	1 (1.3)	0.488
Double mutant (c)	NE**FK**V	0 (0.0)	5 (12.2)	4 (14.8)	9 (11.3)	0.554
Double mutant (d)	NE**FN**V	0 (0.0)	2 (4.9)	0 (0.0)	2 (2.5)	0.650
Triple mutant	NE**FKL**	0 (0.0)	1 (2.4)	0 (0.0)	1 (1.3)	1.000

* Point mutations are shown in boldface. ^#^ Comparison between the three different sites was done by Fisher’s exact test (*p* < 0.05).

**Table 5 genes-10-00692-t005:** Prevalence of multidrug resistance-associated protein 1 (*pfmrp1*) mutations and haplotypes in asymptomatic *P. falciparum* isolates collected from three townships of Myanmar.

Mutation at Codon		Number of Isolate (%)				*p* Value ^#^
		Laiza	Banmauk	Paletwa	Total	
		n = 12	n = 43	n = 27	n = 82	
H191**Y**		9 (75.0)	21 (48.8)	24 (88.9)	54 (65.9)	0.002
N325**S**		1 (8.3)	1 (2.3)	0 (0.0)	2 (2.4)	0.379
S437**A**		9 (75.0)	21 (48.8)	24 (88.9)	54 (65.9)	0.002
F572**L**		1 (8.3)	1 (2.3)	0 (0.0)	2 (2.4)	0.379
H785**N**		7 (58.3)	11 (25.6)	0 (0.0)	18 (22.0)	0.000
I876**V**		8 (66.7)	20 (46.5)	21 (77.8)	49 (59.8)	0.029
T1007**M**		7 (58.3)	13 (30.2)	21 (77.8)	41 (50.0)	0.000
A1339**S**		0 (0.0)	2 (4.7)	0 (0.0)	2 (2.4)	0.650
F1390**I**		1 (8.3)	6 (14.0)	0 (0.0)	7 (8.5)	0.135
**Haplotype**	**Codon** *					
Wild-type	HNSFHITAF	3 (25.0)	20 (46.5)	3 (11.1)	26 (31.7)	0.006
Single mutant	HNSFH**V**TAF	0 (0.0)	2 (4.7)	0 (0.0)	2 (2.4)	0.650
Double mutant	**Y**N**A**FHITAF	0 (0.0)	2 (4.7)	3 (11.1)	5 (6.1)	0.433
Quadruple mutant (a)	**YSA**FH**V**TAF	1 (8.3)	1 (2.3)	0 (0.0)	2 (2.4)	0.379
Quadruple mutant (b)	**Y**N**A**FH**VM**AF	0 (0.0)	1 (2.3)	21 (77.8)	22 (26.8)	0.000
Quadruple mutant (c)	**Y**N**A**FH**V**TA**I**	1 (8.3)	3 (7.0)	0 (0.0)	4 (4.9)	0.357
Quintuple mutant (a)	**Y**N**A**F**NVM**AF	6 (50.0)	10 (23.3)	0 (0.0)	16 (19.5)	0.000
Quintuple mutant (b)	**Y**N**A**FH**VM**A**I**	0 (0.0)	1 (2.3)	0 (0.0)	1 (1.2)	1.000
Quintuple mutant (c)	**Y**N**A**FH**V**T**SI**	0 (0.0)	2 (4.7)	0 (0.0)	2 (2.4)	0.650
Sextuple mutant	**Y**N**ALNVM**AF	1 (8.3)	1 (2.3)	0 (0.0)	2 (2.4)	0.379

* Point mutations are shown in boldface. ^#^ Comparison between the three different sites was done by Fisher’s exact test (*p* < 0.05).

**Table 6 genes-10-00692-t006:** Prevalence of Kelch protein 13 (*k13*) mutations and haplotypes in asymptomatic *P. falciparum* isolates collected in three areas.

Mutation at Codon	Number of Isolate (%)				*p* Value ^e^
	Laiza	Banmauk	Paletwa	Total	
	n = 12	n = 23	n = 22	n = 57	
N insertion ^a^	0 (0.0)	1 (4.3)	0 (0.0)	1 (1.8)	1.000
NN insertion ^b^	12 (100.0)	7 (30.4)	7 (31.8)	26 (45.6)	0.000
NNN insertion ^c^	0 (0.0)	1 (4.3)	3 (13.6)	4 (7.0)	0.423
K191**T**	0 (0.0)	4 (17.4)	0 (0.0)	4 (7.0)	0.067
F446**I**	8 (66.7)	0 (0.0)	0 (0.0)	8 (14.0)	0.000
P574**L**	1 (8.3)	0 (0.0)	0 (0.0)	1 (1.8)	0.211
**Haplotype** ^d^					
Wild-type	0 (0.0)	10 (43.5)	12 (54.5)	22 (38.6)	0.003
N insertion	0 (0.0)	1 (4.3)	0 (0.0)	1 (1.8)	1.000
NN insertion	3 (25.0)	7 (30.4)	7 (31.8)	17 (29.8)	1.000
NNN insertion	0 (0.0)	1 (4.3)	3 (13.6)	4 (7.0)	0.423
NN insertion F446**I**	8 (66.7)	0 (0.0)	0 (0.0)	8 (14.0)	0.000
NN insertion P574**L**	1 (8.3)	0 (0.0)	0 (0.0)	1 (1.8)	0.211
K191**T**	0 (0.0)	4 (17.4)	0 (0.0)	4 (7.0)	0.067

^a^ N insertion between amino acids 136 and 137. ^b^ NN insertion between amino acids 136 and 137. ^c^ NNN insertion between amino acids 136 and 137. ^d^ Point mutations are shown in boldface. ^e^ Comparison between the three different sites was done by Fisher’s exact test (*p* < 0.05).

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
