# Peer review of "Genetic Variations Associated with Drug Resistance Markers in Asymptomatic Plasmodium falciparum Infections in Myanmar"

_genes, 2019, doi:10.3390/genes10090692_

Round 1

Reviewer 1 Report

 Genetic Variations Associated with Drug Resistance  Markers in Asymptomatic Plasmodium falciparum Infections in Myanmar

Summary:  Malaria is still a big problem in Greater Mekong Sub-region of South East Asia. The authors of state that there is limited data on drug resistance in asymptomatic infections in Mynamar area. Using standard PCR and sequencing methods, they determined the genetic variations in drug resistance markers of Plasmodium falciparum parasites obtained from asymptomatic populations in three geographically dispersed townships of Myanmar. They hope the results will  provide updated information on the drug resistance status of P. falciparum malaria in multiple sentinel sites in Myanmar, which is necessary for developing strategies to eliminate P. falciparuminfections in this country. They reported 43 distinct haplotypes potentially associated with multidrug resistance.

Comments:

Overall, the paper is well written, and the methods, which are standard for these kinds of studies, are very clearly explained.

However, I have a few additional comments

Why was haplotype analysis conducted? The rationale behind the haplotype network analysis was explained. A small statement to explaining the rationale behind this analysis would be helpful Sentence 127, in the result section – you have cited references in the results section – this is not the standard way of reporting – you do not need to cite references – state your findings. Rephrase of remove? It is a little strange that all the 12 samples from Laiza had all the pfcrt mutations. Could this have been caused by sample contamination? In the discussion section, there is no mention of malaria or the resistance in symptomatic patients in the area. This could give people a better perspective of the seriousness of the resistance problem in the area.

Author Response

Response to Reviewers’ Comments

We thank the reviewers for their support and constructive comments.

Reviewer #1

Summary:  Malaria is still a big problem in Greater Mekong Sub-region of South East Asia. The authors of state that there is limited data on drug resistance in asymptomatic infections in Myanmar area. Using standard PCR and sequencing methods, they determined the genetic variations in drug resistance markers of Plasmodium falciparum parasites obtained from asymptomatic populations in three geographically dispersed townships of Myanmar. They hope the results will provide updated information on the drug resistance status of P. falciparum malaria in multiple sentinel sites in Myanmar, which is necessary for developing strategies to eliminate P. falciparum infections in this country. They reported 43 distinct haplotypes potentially associated with multidrug resistance.

Overall, the paper is well written, and the methods, which are standard for these kinds of studies, are very clearly explained.

However, I have a few additional comments:

1. Why was haplotype analysis conducted? The rationale behind the haplotype network analysis was explained. A small statement to explaining the rationale behind this analysis would be helpful.

Response: Thanks for the suggestion. The haplotype network helps illustrate the relationship and potential origins of the parasites. Explanation was added accordingly.

2.In the result section – you have cited references– this is not the standard way of reporting – you do not need to cite references – state your findings. Rephrase or remove?

Response: This research used parasite samples from our previous study. To provide the readers some context, we cited these references in the results section.

3. It is a little strange that all the 12 samples from Laiza had all the pfcrt mutations. Could this have been caused by sample contamination?

Response: We do not think that contamination is an issue here, as we had proper negative controls. Furthermore, this result is consistent with earlier analysis of samples collected from the same area, which also showed fixation of the pfcrt mutations (e.g., Zeng W. et al. 2017. Antimicrob Agents Chemother. 61, e01689-16).

4. In the discussion section, there is no mention of malaria or the resistance in symptomatic patients in the area. This could give people a better perspective of the seriousness of the resistance problem in the area.

Response: Thanks for this comment. We now included drug resistance in symptomatic patients on page 9, and discussed the differing prevalence of k13 and pfmdr1 mutations in different regions.

Reviewer 2 Report

This is an interesting study addressed to the long lasting, but still acute and serious, problem of malaria resistance to drugs.  Although focused in Myanmar, it is of general significance and with conclusions applicable to many other sites of the world in which malaria requires a continuous and imaginative fight.  The work uses samples from a large cohort of residents, as well as appropiate analytical approaches, generating a trustable study which might have applicative consequences. The manuscript reads well and had a well equilibrated analysis and treatment of the most significant issues. The analysis of the haplotype network is however one of the issues that might deserve a more extended discussion.

Author Response

Response to Reviewers’ Comments

We thank the reviewers for their support and constructive comments.

Reviewer #2

This is an interesting study addressed to the long lasting, but still acute and serious, problem of malaria resistance to drugs.  Although focused in Myanmar, it is of general significance and with conclusions applicable to many other sites of the world in which malaria requires a continuous and imaginative fight.  The work uses samples from a large cohort of residents, as well as appropriate analytical approaches, generating a trustable study which might have applicative consequences. The manuscript reads well and had a well equilibrated analysis and treatment of the most significant issues. The analysis of the haplotype network is however one of the issues that might deserve a more extended discussion.

Response:Thanks for your approval and the comment. We’ve extended discussion about the haplotype network in the manuscript.

Reviewer 3 Report

In this paper, the authors use a PCR sequencing approach to characterize Plasmodium falciparum (a causative agent of malaria) parasite genes from isolates in Myanmar, a country in Southeast Asia where malaria is endemic. Drug resistance in this region has notoriously been a problem, making investigation into this topic very relevant. Using this sequence data, the authors aims to quantify the prevalence of drug-resistant parasites from several townships in Myanmar. While this is an important topic worthy of reporting and potentially a valuable resource for future studies, there is a rapidly growing literature on this topic taking similar or better approaches in this region, reducing the novelty/originality of the project compared with other published material.

The paper overall is well-written. The tables and figures are very well constructed and ordered, and help the reader understand the results being shown. The text is clear and generally easy to read. However, there is an over-abundance of haplotype names throughout the text of the paper which could be seen in the tables of the results section, making those sections difficult to read. Also, the discussion section seems to be just a second outline of the results with some more information about particular mutations that were found, with little conceptual discussion about what these results could truly add to this subject. I would recommend drastic reduction of the results section due to extensive redundancy with the tables and figures, which would make the paper easier to read.

The conclusions are very consistent with the evidence and arguments presented, and address the main question posed. There was no over-statement of conclusions, and as a purely descriptive report this paper is scientifically very sound. There are no major disagreements of these results with the current academic consensus, so there is not an urgency to to make a substantial case for the validity of their results.

As my first moderate concern, since the deposition of these parasite gene sequences into GenBank are the primary contribution of this study, in my opinion, I would strongly suggest that those accession numbers would be required and added to the paper before publication.

As my second moderate concern, while the PCR sequencing methods section references other studies and is a well-established method, since this is the primary technique on which this entire study is dependent, I would recommend expanding this section to include a discussion of quality control to answer the following questions:

How was the quality of the sample DNA assessed? Which sequencing platform was used? Were duplicate PCR sequencing reactions performed to control for PCR errors? How was sequencing quality incorporated into the analysis, and were any sequences excluded for any reason?

As a very minor concern, for Figure S1 I would incorporate the names of the genes immediately adjacent to the gene diagrams, not only in the figure legend.

Author Response

Response to Reviewers’ Comments

We thank the reviewers for their support and constructive comments.

Reviewer #3

In this paper, the authors use a PCR sequencing approach to characterize Plasmodium falciparum (a causative agent of malaria) parasite genes from isolates in Myanmar, a country in Southeast Asia where malaria is endemic. Drug resistance in this region has notoriously been a problem, making investigation into this topic very relevant. Using this sequence data, the authors aims to quantify the prevalence of drug-resistant parasites from several townships in Myanmar. While this is an important topic worthy of reporting and potentially a valuable resource for future studies, there is a rapidly growing literature on this topic taking similar or better approaches in this region, reducing the novelty/originality of the project compared with other published material.

The paper overall is well-written. The tables and figures are very well constructed and ordered, and help the reader understand the results being shown. The text is clear and generally easy to read.

However, there is an over-abundance of haplotype names throughout the text of the paper which could be seen in the tables of the results section, making those sections difficult to read. Also, the discussion section seems to be just a second outline of the results with some more information about particular mutations that were found, with little conceptual discussion about what these results could truly add to this subject. I would recommend drastic reduction of the results section due to extensive redundancy with the tables and figures, which would make the paper easier to read.

Response:Thanks for your suggestion. We have simplified the haplotype names and reduced the result section. We also revised the discussion section accordingly.

As my first moderate concern, since the deposition of these parasite gene sequences into GenBank are the primary contribution of this study, in my opinion, I would strongly suggest that those accession numbers would be required and added to the paper before publication.

Response:We are uploading our sequences to GenBank for the accession numbers.

As my second moderate concern, while the PCR sequencing methods section references other studies and is a well-established method, since this is the primary technique on which this entire study is dependent, I would recommend expanding this section to include a discussion of quality control to answer the following questions:

How was the quality of the sample DNA assessed? Which sequencing platform was used? Were duplicate PCR sequencing reactions performed to control for PCR errors? How was sequencing quality incorporated into the analysis, and were any sequences excluded for any reason?

Response:We used the NanoDrop 2000C to assess the quality of DNA samples. Samples that were used for PCR-sequencing typically had a OD260/280 ratio of 1.6 - 1.8 and DNA concentration of >5 ng/uL. PCR amplicons were purified and sequenced in an ABI 3730XL DNA analyzer. We evaluated the quality of sequencing by examining the chromatograms. To ensure sequence quality, all DNA fragments were sequenced for both strands.

As a very minor concern, for Figure S1 I would incorporate the names of the genes immediately adjacent to the gene diagrams, not only in the figure legend.

Response:According to suggestion, we have revised the figure.
